# A Combined mmWave Tracking and Classification Framework Using a Camera for Labeling and Supervised Learning

**DOI:** 10.3390/s22228859

**Published:** 2022-11-16

**Authors:** Andre Pearce, J. Andrew Zhang, Richard Xu

**Affiliations:** Global Big Data Technologies Center, School of Electrical and Data Engineering, University of Technology Sydney, Sydney 2007, Australia

**Keywords:** mmWave, sensing, fusion, automated labeling

## Abstract

Millimeter wave (mmWave) radar poses prosperous opportunities surrounding multiple-object tracking and sensing as a unified system. One of the most challenging aspects of exploiting sensing opportunities with mmWave radar is the labeling of mmWave data so that, in turn, a respective model can be designed to achieve the desired tracking and sensing goals. The labeling of mmWave datasets usually involves a domain expert manually associating radar frames with key events of interest. This is a laborious means of labeling mmWave data. This paper presents a framework for training a mmWave radar with a camera as a means of labeling the data and supervising the radar model. The methodology presented in this paper is compared and assessed against existing frameworks that aim to achieve a similar goal. The practicality of the proposed framework is demonstrated through experimentation in varying environmental conditions. The proposed framework is applied to design a mmWave multi-object tracking system that is additionally capable of classifying individual human motion patterns, such as running, walking, and falling. The experimental findings demonstrate a reliably trained radar model that uses a camera for labeling and supervision that can consistently produce high classification accuracy across environments beyond those in which the model was trained against. The research presented in this paper provides a foundation for future research in unified tracking and sensing systems by alleviating the labeling and training challenges associated with designing a mmWave classification model.

## 1. Introduction

The process of training millimeter wave (mmWave) sensors to solve classification problems is rapidly becoming more popular and proving to be a promising direction in radar sensing research. Some of the most promising techniques that are being pursued in this field of research include deep-learning-based approaches. However, successfully using a deep-learning-based approach typically requires an abundant set of training data to adequately teach a model the relevant features that can be relied on for classification. Constructing a large and meaningful dataset requires a domain expert to spend time appropriately labeling the raw data collected from the sensor. This process can be quite troublesome, specifically when dealing with mmWave raw data, which is notoriously difficult to correctly intuitively label.

To solve this challenge, one potential direction is through information fusion—more specifically, the fusion of mmWave radar and camera. As a result, it is important to understand the processes involved in general information fusion with respect to mmWave radar and camera. Information fusion with mmWave radar and camera refers to the combination of the two independent streams of data so that they are presented and interpreted in a unified perspective [1]. There are a number of different variables that are involved in achieving this fused state of information. In an attempt to break down the varying components involved in information fusion [2], the following high-level characteristics should be considered:System architecture.Fusion depth.Fusion process.Fusion algorithm.

The system architecture of mmWave radar and camera fusion focuses on the high-level structure that the fusion process operates on. In a review article presented by [2], the authors identified three major fusion structures that are commonly abstracted in related literature. These three types of fusion system architectures, along with their respective benefits and limitations, are depicted in Table 1.

The three types of fusion architectures presented in Table 1 ultimately describe the major architecture types found in the existing research. The rationale responsible for deciding which architecture type to implement over the others fundamentally stems from the run-time requirements that a given solution must meet. The next characteristic that can be used to describe mmWave radar and camera information fusion is the depth of the fusion that is performed.

The authors of [3,4] termed this characteristic as the level of fusion. This simply refers to the point in which the mmWave data is fused with the camera data, starting from the primitive point in which raw data is collected and stemming until a point where fusion might take place only once several layers of processing have already taken place independently for both/either the radar and/or the camera.

The authors of [3,4] abstracted these depths of fusion into three progressive levels. These levels are further described in Table 2.

The fusion process is another aspect that can differentiate the fusion that takes place for mmWave radar and a camera. The fusion process ultimately refers to the basis upon which the actual fusion of the two sensors takes place. There are a number of different approaches that can serve as the means to perform fusion. One method explored and demonstrated by the authors of [5] attempts to spatially fuse the mmWave radar and camera. This process refers to the mmWave radar and camera each recording data in their own coordinate system. Following this, each of the sensor’s coordinate systems should be transformed into a world coordinate system that closely depicts the three-dimensional coordinate system via which we perceive the world.

Another fusion process that is closely related to spatial fusion—and perhaps necessary for spatial fusion to take place—is fusion through sensor calibration. There are a number of varying techniques presented for calibrating mmWave and camera sensors, such as the work presented by the authors of [6,7,8,9]. Lastly, the most simple process in which the basis of fusion can take place is temporally. Regardless of the basis in which the fusion takes place, an appropriate correlation and association algorithm needs to be designed and implemented.

The research discussed in this paper presents a framework for automated labeling of mmWave radar data using information fusion theory with a camera. The research and methodologies that we propose in this paper are novel in three major respects. First, the generalized automated labeling framework that we present is one of the first proposed in the context of mmWave, where an attempt has been made to abstract the specific teacher and student objectives from the framework. Secondly, the framework that we present is also one of the first of its kind to encompass the complete processing chain for training a standalone mmWave radar classification model using a camera as a teacher. Lastly, the example implementation of the framework that we present demonstrates a novel adaption for the correlation and fusion of camera and radar data. These primary contributions that we present are further explained in the following:The radar training with a camera labeling framework that we present is generalized by definition as it is not specific to a given classification problem in either the radar or camera domain. The agnostic nature of the framework we propose is the first of its kind (that we are aware of) in the context of mmWave radar. Existing approaches are either specific to the task of object detection or specific to the classification problem that the given authors are attempting to solve.The framework that we present in Section 2.1 is the first of its kind that includes a suggested approach towards all stages in the processing chain involved in achieving a radar classifier. Existing approaches usually have a focus on presenting a framework that only shows a means for labeling camera data, usually specific to the task at hand, and applying it to either raw or pre-processed radar data. Our framework also satisfies that objective but takes the labeled data further and demonstrates how this labeled radar data can be used in a teacher-and-student-based approach to form a standalone radar classifier.To demonstrate the feasibility of the framework proposed, we also demonstrate a practical implementation of our proposed framework. In our example implementation, we demonstrate how a pre-trained camera classifier can be used to label raw mmWave data for human activity recognition in conjunction with performing mmWave multiple-object tracking. The correlation technique that we devised and utilized is unique and a looser form of the calibration that takes place between the camera and radar. This removes the need for tight coupling between raw radar points and points in the vision domain.

The remainder of this paper will be structured as follows. First, we will continue to explore the related literature to gain a deeper understanding of the existing frameworks that solve similar problems. Secondly, we will discuss the problem space that we aim to propose a viable solution for. Thirdly, the proposed approach will be introduced, and the methodology surrounding our efforts will be detailed and rationalized. Fourthly, using the methodology we present, an implementation of the proposed framework will be discussed to ultimately serve as a demonstration of the practicality of the framework. Lastly, we will discuss the outcome of utilizing the framework we present with respect to the aforementioned example implementation.

### 1.1. Review of Related Literature

As new deep-learning-based sensing research is being released for mmWave radar, difficulties associated with the labeling of mmWave data are increasingly being identified. As a result, different labeling strategies have been presented in recent literature, ultimately demonstrating the feasibility of using another sensor, such as cameras, to label datasets collected by radar.

One of the earlier pieces of research that demonstrated a fusion-based approach with radar and a camera to classify objects is the work presented by the authors of [10]. The authors of [10] deconstructed the problem into a two-stage approach. The first stage involves recording the data and performing a typical Kalman-filter-based approach to identify objects in the field of view of the radar. The last stage involves taking the points of interest identified in the radar data and projecting those points onto the same plane as the camera. The purpose of this is to highlight points of interest that a deep classification can be performed on.

Another more recent piece of literature that demonstrates an approach to the fusion of mmWave radar and camera is the work presented by [11]. The approach discussed by the authors of [11] is largely similar to the technique presented in [10]. The authors of [11] proposed a method that jointly uses radar and camera to detect objects. Similar to [10], the initial object detection is performed by filtering the mmWave radar data. Following this, the mmWave data is projected onto the image plane through coordinate translation via camera calibration. Finally, using this combined state, machine learning is used to identify and track the objects in the field of view.

Some more recent works, presented by the authors of [12], produced a labeled Frequency-Modulated Continuous-Wave (FMCW) dataset with correlated inertial measurement unit measurements and corresponding camera frames. The labeling strategy proposed by the authors of [12] ultimately relies on time synchronization between the three sensors. After temporally aligning the sensors, the authors required spatial calibration between the radar and camera in order to match the detected objects. The technique used in [12] to spatial calibrate the radar and camera involves introducing an object that is both distinctly identifiable in vision and reflective in the radar domain. This object is used to induce a strong reflective point in the radar’s heatmap, which can ultimately be used as a reference point for the corresponding space in the camera’s domain.

The work of [12] leads to an interesting question regarding the techniques available to calibrate a mmWave radar with camera sensors. A review presented by [13] breaks this question down into three overarching components that encompass sensor calibration in the context of radar and vision fusion as presented in modern literature:Coordinate calibration—the alignment of individual points in the radar with objects in camera’s field of view. This initial stage of calibration can be seen implemented in three varying mechanisms within the works presented by [14,15,16].Radar point filtering—where noise and undesirable data is acknowledged and filtered from the radar data. The work of [17] presents an approach that demonstrates calibration involving the filtering of undesired data points based on speed and angular velocity.Error calibration—refers to the processes implemented to overcome errors in the calibrated data. There are many methods that have been devised to attempt to overcome calibration error. One approach presented by [18] demonstrates an Extended Kalman Filter that is used to model the measurement errors present in the independent sensors.

The authors of [19,20] proposed two similar approaches that demonstrated object detection through the fusion of radar and camera. Both of the techniques demonstrated an Artificial Neural Network (ANN), where the inputs are pre-processed radar data and raw camera data. The primary difference between the two techniques is that the authors of [19] pre-processed the radar data to produce range-azimuth images as an input for the ANN, while the authors of [20] pre-processed the radar data to form 2D point-cloud data and utilized this as the input into the ANN.

Lastly, an approach presented by the authors of [21] demonstrated an auto-labeling framework, achieving a similar goal to what we present in our paper but through different means. The approach presented by [21] uses an active learning system based on a Convolutional Neural Network (CNN). Although the technique presented by [21] demonstrated promising results, it is important to note that the technique requires human input to manually label ambiguous data. The framework that we present in this paper demonstrates an approach that requires no human interaction for the labeling of radar frames.

## 2. Materials and Methods

The purpose of this section is to explore the methodology behind the framework that we present in our research and demonstrates its practicality. As such, this section of the paper has been divided into two parts. The first part, Section 2.1, details the framework itself, including the stages and components involved. The second part, Section 2.2, takes the framework presented in Section 2.1 and demonstrates how it can be practically applied to a problem.

### 2.1. Radar Training with Camera Labeling and a Supervision Methodology

This section of the paper describes and illustrates a generalized methodology for labeling radar data and training a standalone radar model using a camera as the ground truth for the radar model. The purpose of this methodology is to provide a framework for others to follow when attempting to extend camera-based models into a radar-based model. The methodology described in this section is practically applied and demonstrated in Section 2.2 of this paper.

#### 2.1.1. Problem Space

Raw radar data is notoriously difficult to intuitively interpret without applying pre-processing techniques to extract the desired information. Furthermore, the labeling of raw radar data can be a difficult and tedious task even for a domain expert. This is usually due to the large dataset sizes that are involved. As a result of this labeling difficulty, training a model that utilizes radar data to classify complex events also becomes a difficult task.

This problem is typically addressed in the existing literature by reducing the dataset size of the radar data or by restricting the potential of the classifier being trained to only a small set of classification types. Although this may alleviate the problem, there are negative implications to the potential of the designed radar model. Therefore, there is an evident need to devise a solution to simplify the labeling approach for radar data that, in turn, can be utilized to train a classifier without impacting the constraints of the designed model.

Camera classification networks are a well-defined and researched domain. As seen in Section 1.1 of this paper, there are many existing models available that demonstrate successful classification capabilities for a variety of complex movements. The methodology proposed in this paper uses a camera as a means of addressing the labeling challenge with raw radar data and the inherit training difficulty of standalone models/classifier networks. Attempting to ultimately use vision data to label and act as the ground truth for radar data presents two major challenges that need to be considered.

First, vision data is inherently a snapshot of a horizontal and elevation domain at a given point in time. In other words, the perspective of two dimensional data is typically considered to be still/static in nature. Radar data, on the other hand, is typically a perspective of a range/distance and relative angle or of an inferred horizontal plane. Additionally, radar data in this domain is also typically collected on moving/dynamic objects. This domain-alignment issue, between camera and radar data, ultimately poses as a challenge around the correlation of static objects present in vision data with moving objects present in the radar data.

The second major challenge identified is also a correlation problem by nature, presenting itself when operating in an environment where multiple objects are simultaneously present and/or moving in the field of view. This scenario ultimately surfaces the challenge of correctly associating multiple objects in the vision data with the same objects in the radar data.

#### 2.1.2. Proposed Approach

This section depicts the proposed solution methodology to the previously discussed problem space. The methodology proposed in this paper should be interpreted as a framework that can be applied to a given camera classification model so that radar can achieve an ideally equally performing standalone classification network.

The proposed approach can be conceptually considered in the following three stages:Data collection.Correlation and labeling.Radar training.

Figure 1 illustrates the generalized processing chain that is involved throughout the aforementioned three high-level stages. The data collection stage is an abstraction in the framework that is responsible for collecting data independently from the radar and camera. The data collected from each of the different sensors is then under-taken through the appropriate pre-processing and normalization methods depending on the particular application that this framework is being applied to. The desired output state, for the radar data, is a sequence of radar data frames across the time domain. At this stage, the camera data should be in a state that is consumable by the camera classifier network that is being applied to train the radar with.

After successful data collection and the appropriate transformations, the pre-processed camera data should then be applied to the camera classifier that is being implemented to train the radar. The expectation of the camera classifier is to perform the respective classifications against the camera data so that a sequence of camera frames with labeled classifications can be obtained. The domain in which these camera frames are obtained can be considered abstract for the definition of this methodology as this is dependent on the particular application.

An important part of the proposed methodology is the correlation approach to synchronize the camera and radar data. The time associated with the sample taken for the camera data is used as a reference so that the radar data can be extrapolated in order to synchronize with time relative to the camera. Figure 2 demonstrates the time bias that is present between the radar and camera samples.

Assuming two consecutive radar time stamps are expressed as $S_{r}\left(n-2\right)$ and $S_{r}\left(n-1\right)$, the next sample predicted by the radar can be expressed as $S_{f}\left(n\right)$. Using the position and velocity components of the radar data, $S_{r}\left(n-1\right)$ and $S_{f}\left(n\right)$ can be linearly interpolated to estimate the radar sample at the correlating camera data point $S_{e}\left(n\right)$. This process ultimately satisfies Equation (Equation 1).
(1)$$k=\frac{\Delta S_{e}\left(n\right)}{\Delta t}$$

Once both radar and camera data has been correlated using the above approach, a labeled set of radar frames can be formed based on the correlation that was achieved. The labeled set of frames $F_{l}\left(n\right)$ can then be subjected to training for classification of the desired feature sets encoded in the radar and camera data.

The classification network applied to the labeled radar data can be an abstracted problem in the context of the framework proposed in this paper. The proposed approach ultimately abstracts the design challenges associated with fusing and labeling the radar data with camera classifiers. As a result, a generic model can be applied and trained against the labeled radar frames, based on the original camera classification network that was selected. The next section of this paper demonstrates a practical implementation of the generalized methodology illustrated.

### 2.2. System Design and Implementation

This section demonstrates an implementation of the labeling and supervision framework presented in the previous section. As mentioned in Section 2.1, the framework provides a means to training radar using labeled camera frames. As such, the design and implementation discussed in this section demonstrates the framework’s suitability by applying it to train a mmWave tracking system to classify human movement patterns.

Figure 3 illustrates the overall system design that implements the framework that is discussed in Section 2.1 and illustrated in Figure 1. The system design presented contains three high-level processing pipelines:Radar Pipeline.Camera Pipeline.Fused Pipeline.

The remainder of this section will continue to break down the system design with respect to each of the pipelines illustrated in Figure 3.

#### 2.2.1. Radar Pipeline

The radar pipeline is associated with the processing required to prepare the radar data for fusion with the camera frames. As seen in Figure 3, it is expected that the radar pipeline can achieve object detection and tracking. Figure 4 further extends the high-level aspects of the radar pipeline presented in Figure 3.

The radar-processing pipeline has been broken down into four different sub-modules. The *Radar Data Collection* module is responsible for collecting the raw analog-to-digital converter (ADC) data from the radar. The ADC used was the integrated ADC on the Texas Instruments IWR6843 mmWave radar, configured to operate at a resolution of 16 bits. The ADC was set to sample at 10 mega-samples per second (Msps), collecting a total of 256 samples per chirp cycle. The raw radar data is then processed to perform two fast Fourier transformations (FFT), the range FFT followed by the Doppler FFT. These transformations are necessary so that the respective range-Doppler heatmaps can be generated for each radar frame. An example range-Doppler heatmap can be seen in Figure 5.

The second module of the radar-processing pipeline is the *Constant False Alarm Rate (CFAR)* stage, which is ultimately responsible for implementing a CFAR filter for performing object detection on the range-Doppler heatmaps. It is important to note the decision to operate with range-Doppler heatmaps as this decision was made primarily for the later radar classifications that will be discussed.

Following the object detection in the range-Doppler heatmaps, the data is further processed to be illustrated as point-cloud data so that traditional radar point cloud clustering and tracking can take place using density-based spatial clustering of applications with noise (DBSCAN) and a Kalman filter. The radar hardware architecture used in this system was a Texas Instruments IWR6843 mmWave radar with a DCA1000EVM for capturing the raw ADC data of the radar.

#### 2.2.2. Camera Pipeline

The camera pipeline is responsible for preparing and labeling the camera frames for fusion with the radar range-Doppler heatmaps. The data that is recorded from the camera must first be processed for object detection and each object coordinate that is mapped in the field of view. Following this, the appropriate movement classifications can be made and associated with objects in the field of view of the camera. Figure 6 illustrates a granular perspective of the stages involved in the camera processing pipeline.

As illustrated in Figure 6, object detection is the first task that is performed in the camera processing pipeline. In order to realize camera object detection, a Faster Region-Based Convolutional Neural Network (Faster R-CNN) is implemented. The structure implemented can be seen in Figure 7 and is based on the research presented in [22].

The generalized loss function adopted for camera object detection follows the multi-task loss in Fast R-CNN [23]. The loss equation is expressed as follows:(2)$$\begin{array}{c} L\left(\left\{p_{i}\right\},\left\{t_{i}\right\}\right)=\frac{1}{N_{cls}}\sum_{i}L_{cls}\left(p_{i},p_{i}^{*}\right) \end{array}$$(3)$$\begin{array}{c} +\lambda \frac{1}{N_{reg}}\sum_{i}p_{i}^{*}L_{reg}\left(t_{i},t_{i}^{*}\right) \end{array}$$
where *i* refers to the index of an anchor (noting the definition of an anchor as per [22]), $p_{i}$ is the predicted probability of anchor *i* being a detected object, $p_{i}^{*}$ is the ground truth label for the given anchor *i* (derived as per Equation (Equation 4)), $t_{i}$ is the coordinate vector associated with the bounding box of the predicted anchor *i*, and $t_{i}^{*}$ is the ground truth of the bounding box coordinate vector associated with anchor *i* that is an object.

The ground truth label $p_{i}^{*}$, for a given anchor *i*, is binary in value and follows the below expression:(4)$$p_{i}^{*}=\left\{\begin{array}{cc} 1, & \mathrm{anchor}\;i\;\mathrm{is}\;\mathrm{positive}\\ 0, & \mathrm{otherwise} \end{array}\right.$$

Furthermore, the terms $L_{cls}$ and $L_{reg}$ refer to the loss functions for the classifier and regressor, respectively, illustrated in Figure 7. $N_{cls}$ and $N_{reg}$ are the normalization of these two terms. The loss function used for the classifier is as follows:(5)$$L_{cls}\left(p_{i},p_{i}^{*}\right)=-\frac{1}{N}\sum_{i}^{N}p_{i}\log\left(p_{i}\right)+\left(1-p_{i}\right)\log\left(1-p_{i}\right)$$

The loss function used for the regressor is as follows:(6)$$L_{reg}\left(t_{i},t_{i}^{*}\right)=smooth_{L1}\left(t_{i},t_{i}^{*}\right)$$
where the $smooth_{L1}$ function is defined as per [23].

After object detection has been performed, the classification model is then applied to the cropped detected objects. The purpose of the classification model in this implementation is to:Formulate a 2D skeleton for each detected object in the field of view.Classify the human activity that is occurring using the 2D skeleton.

In order to achieve this, each of the detected objects is run through AlphaPose [24] to generate the respective 2D skeleton for the detected object. The result of the AlphaPose system is then passed as an image to a CNN that has been pre-trained to classify poses that are associated with:Walking.Running.Falling.

The pre-trained model is ensured to have an accuracy greater than 92%. The accuracy of this classifier network is important, as it will ultimately be built into the mmWave classification network during the fusion pipeline. In parallel with the classification of the detected objects, their location in the field of view is also jointly estimated using camera calibration. This ultimately results in each detected object *j* having a respective given coordinate $\left(X_{jk},Y_{jk}\right)$ for each camera frame *k*, where *X* is used to denote the horizontal coordinate and *Y* is used to represent the estimated range of the object (as opposed to height). Finally, a Kalman filter is applied to more accurately predict the detected object’s true location whilst being tracked in the field of view.

#### 2.2.3. Fused Pipeline

The fusion pipeline is then finally responsible for associating the tracked objects in the radar domain with the tracked and classified objects in the camera domain. As mentioned in Section 2.1 of this paper, before fusing the two domains, a time bias between the domain samples needs to be accommodated. In our implementation, this is achieved by granulating the radar samples so that positional estimates are calculated between radar samples. These positional estimates are deduced so that they correspond with the sampling rate of the camera system.

The association and correlation of the detected objects is then made so that the tracked objects in the camera domain can be related to the tracked objects in the radar domain. This correlation is made using the deltas of the velocity and acceleration between the respective predicted locations of the camera and radar tracking algorithms. For both the camera and radar, the displacement vector is used for correlation using Pearson’s Correlation Coefficient. This approach consequently removes any detected objects that are not commonly identified across domains, ultimately taking care of the scenario where one sensor picks up an object that the other does not. The displacement vectors, for both camera and radar, are expressed in Equations (Equation 7) and (Equation 8), respectively.
(7)$${\vec{CP}}_{l}=\left[cp_{n}-cp_{n-1},cp_{n-1}-cp_{n-2},\cdots ,cp_{2}-cp_{1}\right]$$
(8)$${\vec{RP}}_{m}=\left[rp_{n}-rp_{n-1},rp_{n-1}-rp_{n-2},\cdots ,rp_{2}-rp_{1}\right]$$
where ${\vec{CP}}_{l}$ and ${\vec{RP}}_{m}$ are the displacement vectors for the camera and radar, respectively, each for a given camera detected object *l* and radar-detected object *m*. For the given detected object, the delta between all camera positional estimates cp in a sliding sample window *n* is calculated. The same is applied to the given detected radar object and its radar positional estimates rp in the sliding sample window *n*.

Using the displacement vectors for camera and radar in Equations (Equation 7) and (Equation 8), the Pearson Correlation Coefficient is calculated for each pair of detected objects in both the camera and radar domains as seen in Equation (Equation 9).
(9)$$r_{lm}=\frac{n\sum {\vec{CP}}_{l}{\vec{RP}}_{m}-\left(\sum {\vec{CP}}_{l}\right)\left(\sum {\vec{RP}}_{m}\right)}{\sqrt{\left[n\sum {\vec{CP}}_{l}^{2}-{\left(\sum {\vec{CP}}_{l}\right)}^{2}\right]\left[n\sum {\vec{RP}}_{m}^{2}-{\left(\sum {\vec{RP}}_{m}\right)}^{2}\right]}}$$
where *r* is computed for all combinations of *l* and *m*. The absolute Pearson Correlation Coefficient $\left|r_{lm}\right|$ is taken, and the maximal *l* and *m* combination is deemed to be the correctly correlated pair.

After correlation of the radar and camera domains, we ultimately have a labeled dataset that we can use to train a model for classification in the radar domain. The structure of the model used for classification of the radar data is a CNN with the input shape pertaining to clustered point-cloud data for a single detected object.

## 3. Results

The system described in Section 2.2 was experimentally tested in varying environmental conditions to prove its performance. The first task was to collect the necessary dataset that can be used to train the radar. The dataset compiled needs to jointly have both camera and radar samples so that the respective data fusion can take place. This cannot be collected independently.

A dataset containing 1000 images was collected across four different sessions, where each session had a different external environment. Two of the sessions were recorded indoors, and the other two were in an outdoor setting. In all recorded sessions, we ensured that we recorded situations that included:No targets in the field of view.A single target in the field of view.Multiple targets in the field of view.

Additionally, the four types of activities were distributed along the 1000 images as per Table 3. The frequency in which these activities took place is not a factor of the 1000 images taken. This is due to the fact that one or more activities could be present several times in a single image. This is a result of the potential for multiple objects to be detected and independently processed in a single frame.

The total dataset and inner classifications were equally shuffled to prevent a bias of randomization between classification types. The shuffled dataset was then divided into training, validation, and testing subsets. The first 60% of the equally randomized recorded dataset was reserved exclusively for the training of the camera classifier and, subsequently, the radar classifier. The next 20% was then used for validation of the trained models, allowing us to further refine the classifiers using the validation dataset. Lastly, once the best performance was obtained, the classifiers were tested against the final reserved 20% of the dataset.

The accuracy results of our final trained radar system are presented in Figure 8. The camera-trained radar classifier is compared with the accuracy of the trained standalone camera system and the manually labeled radar classifier in varying environmental setups. To clarify, each of the aforementioned systems is further described as:**Camera-Trained Radar Classifier:** A radar classifier trained using camera-labeled data via the framework proposed in this paper.**Trained Standalone Camera System:** A camera classifier that is used to label the frames for the camera trained standalone radar classifier.**Manually Labeled Radar Classifier:** A radar classifier, of the same design as the camera trained standalone radar classifier that was trained using manually labeled radar data.

In Figure 8, the radar classifier that was trained using camera-labeled data produced an outcome similar—and, in some circumstances, more superior—to that of the standalone camera classifier. In most “normal” scenarios, the radar classifier performed largely identical to the camera classifier. However, there are two environmental changes that should be noted as outliers.

The first is objects that are distant. In the scenario where the camera-trained radar classifier was attempted with targets at a distance greater than six meters, the accuracy of the model was 7.66% less, compared to the camera classifier. On further analysis of the results, it appears that this is likely due to the fact that the point-cloud data per cluster (i.e., detected object) is much leaner compared with objects that are within 6 m of the radar. The leaner point-cloud data results in a lack of distinguishing features between activities in the radar domain. This challenge could potentially be overcome through some additional design considerations with the chirp of the radar.

The second outlier that is worth noting is the experiment performed in an indoor room with low levels of light. As expected, the camera-trained radar classifier was not impacted by the lighting conditions and, as a result, demonstrates an accuracy that is 56.84% higher than the standalone camera classifier in the same lighting conditions.

Given that the radar was trained using camera-labeled data, the best network we could theoretically achieve with the radar is one that is of equal performance to the teacher network (the standalone camera system). The exceptions to this are any sensor specific characteristics that might inhibit the performance of a given sensor, such as ambient lighting in the context of the camera. This particular regard was evident in the second outlier identified, where the camera-trained radar network performed better than the standalone camera network, simply due to ambient lighting.

Whilst acknowledging the aforementioned outliers, it is evident that the camera-trained radar system performed with a high degree of similarity to the standalone camera system. The trained similarity between the two systems is summarized in Table 4, where a ↓ implies an inferior similarity and an ↑ implies a superior similarity, with respect to the trained radar system. The high degree of similarity between the teacher network (the standalone camera system) and the student network (the camera-trained radar system) demonstrates the suitability of the proposed generalized framework toward training a radar model with camera-labeled data.

In order to better understand the theoretical potential of the radar classifier, the camera-trained radar classifier was compared against a manually labeled radar classifier. The purpose of this comparison scheme was to demonstrate the potential of the implemented radar classifier. The significance of this experiment is to, first, highlight the capability that could potentially be expected with the design of the radar classifier and, secondly, to gain an understanding of the pre-encoded errors that the camera-trained radar classifier incurs as a result of labeling errors in the camera domain.

Figure 9 illustrates the accuracy of the manually labeled trained radar system in contrast to the camera-trained radar system. Figure 9 highlights the theoretical potential that the radar classifier can achieve when trained with a manually labeled dataset. Assuming that the manually labeled dataset is not incorrectly labeled, it is expected that training the radar system with a manually labeled dataset will yield higher results than a camera-trained radar system. This is ultimately due to the fact that a camera-trained radar classifier will incur labeling errors associated with the camera classifier.

This hypothesis is ultimately supported by the results presented in Figure 9. It can be seen that the camera-trained radar system does not meet the same performance as the manually labeled radar system. Despite the theoretical potential of the radar classifier, the performance of the camera-trained radar system implemented using the proposed framework was, on average, 96.52% as good as the camera classifier, as seen in Table 4, when negating the outperforming low-level lighting environment.

## 4. Conclusions

The research presented in this paper demonstrated a framework for developing a classifier for mmWave radars using a camera as a teacher for the mmWave radar student network. The example implementation, presented in Section 2.2, showed how the framework can be implemented to achieve a radar classifier that is as accurate as the teacher camera classifier. This performance was demonstrated without compromising the beneficial characteristics of the radar, such as the non-sensitivity to illumination.

The proposed camera-trained method achieved a level of performance that approached the manually labeled radar system, particularly in cases where the camera could generate accurate recognition performance. Hence, using the proposed framework can provide a significant decrease in the amount of manual labeling needed for radar data. The performance of the camera-trained method was degraded where camera’s recognition was limited. This was specifically seen in the results presented for the “Outdoors with distant objects” environment. In order to further the research presented in this paper, additional camera-based labeling networks should be analyzed, through the methodology presented in Section 2.1, for their ability to train an equally performing radar network.

Furthermore, it would be of interest in future research to conduct radar classifier design optimizations and compare the network performance across a variety of different radar hardware. Performing such an experiment will allow us to better understand the impact of intrinsic radar characteristics, such as the ADC sampling rate and maximum resolution, on the generalized performance of the proposed framework.

The framework presented in Section 2.1 should be considered as a foundation to designing mmWave classifiers. Adopting the framework presented in this paper can help researchers to alleviate the burden associated with the labeling of mmWave data. This labeling challenge usually results in researchers under-collecting an adequate set of training data to design an mmWave classifier. In this scenario, due to the limited training dataset collected, the classification network being designed may not reach its full potential, simply as a result of being deprived of training data. Hence, the framework that we present may assist future research by providing a model that researchers can follow to remove the need for the manual labeling of data when designing a classifier for mmWave radar.

## Figures and Tables

**Figure 1 sensors-22-08859-f001:**
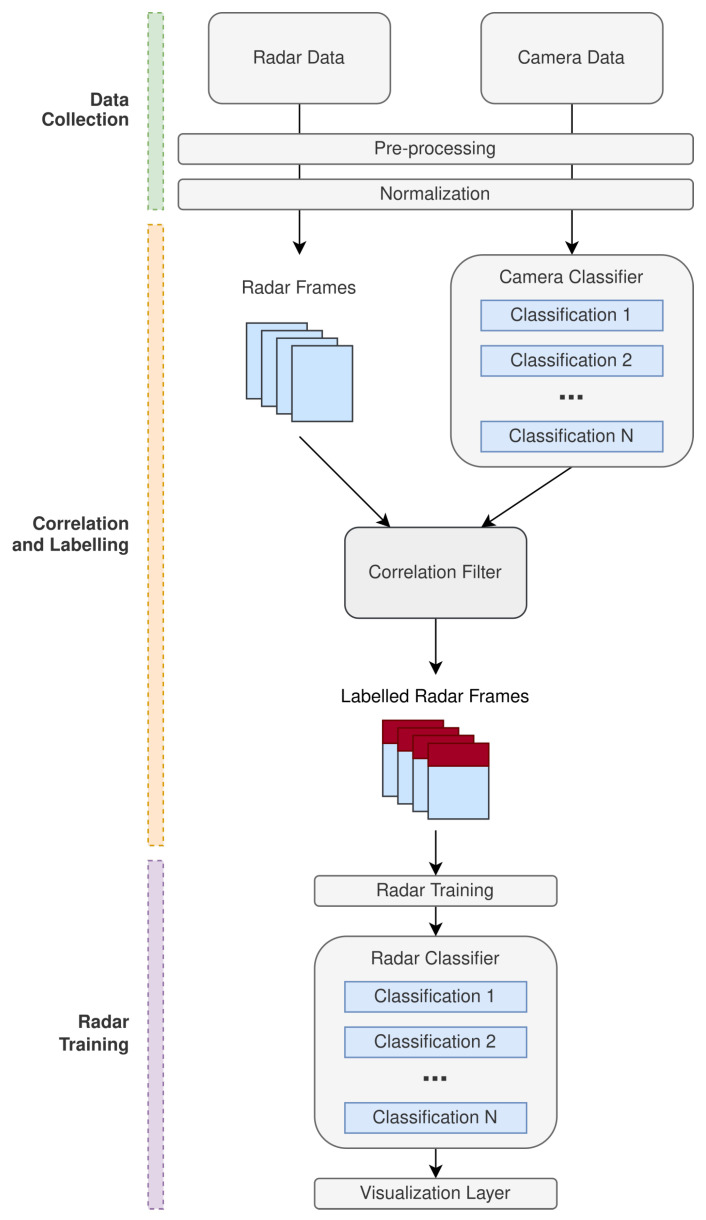
Processing chain for the proposed radar training with camera labeling and supervision methodology.

**Figure 2 sensors-22-08859-f002:**
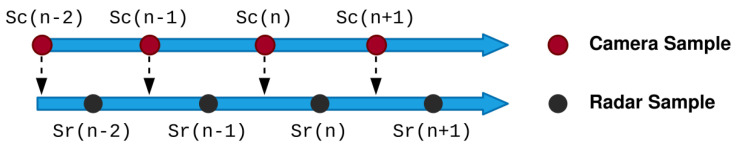
Radar and camera time-alignment bias.

**Figure 3 sensors-22-08859-f003:**
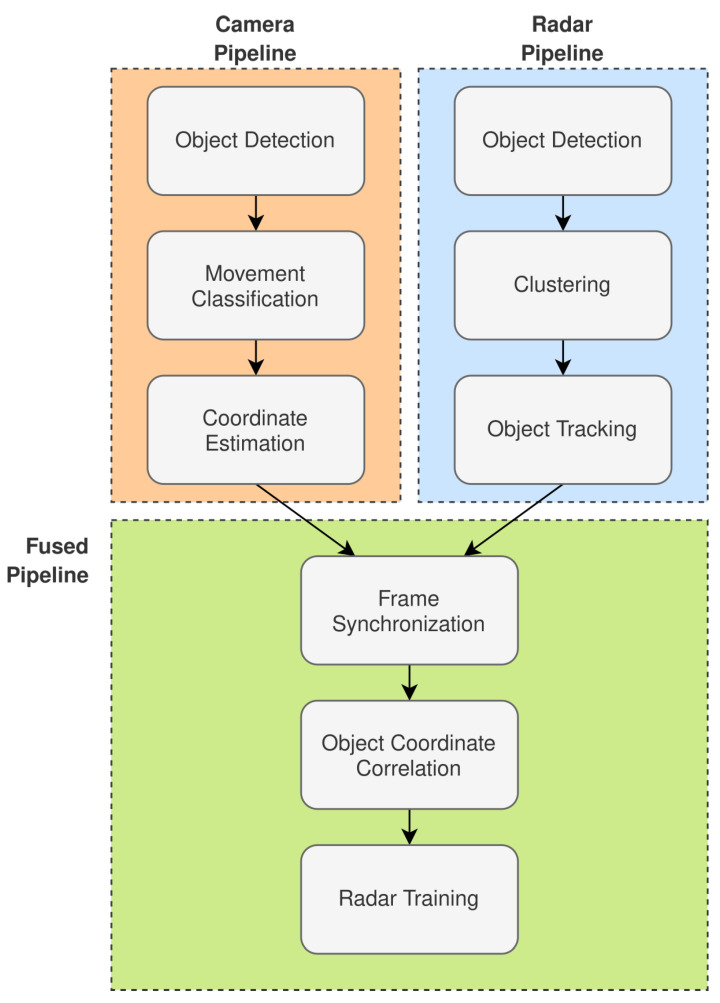
Radar-tracking system design with a human-movement-pattern classifier trained with camera-labeled frames.

**Figure 4 sensors-22-08859-f004:**
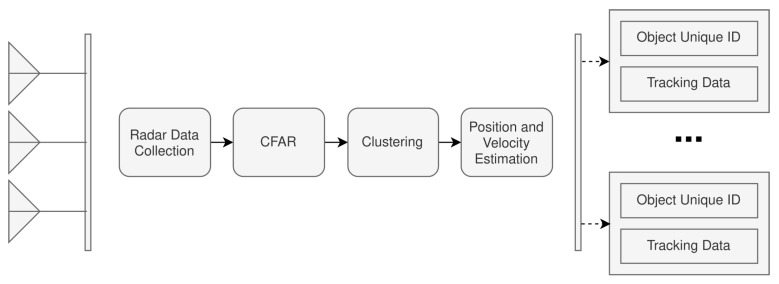
Radar-processing pipeline design.

**Figure 5 sensors-22-08859-f005:**
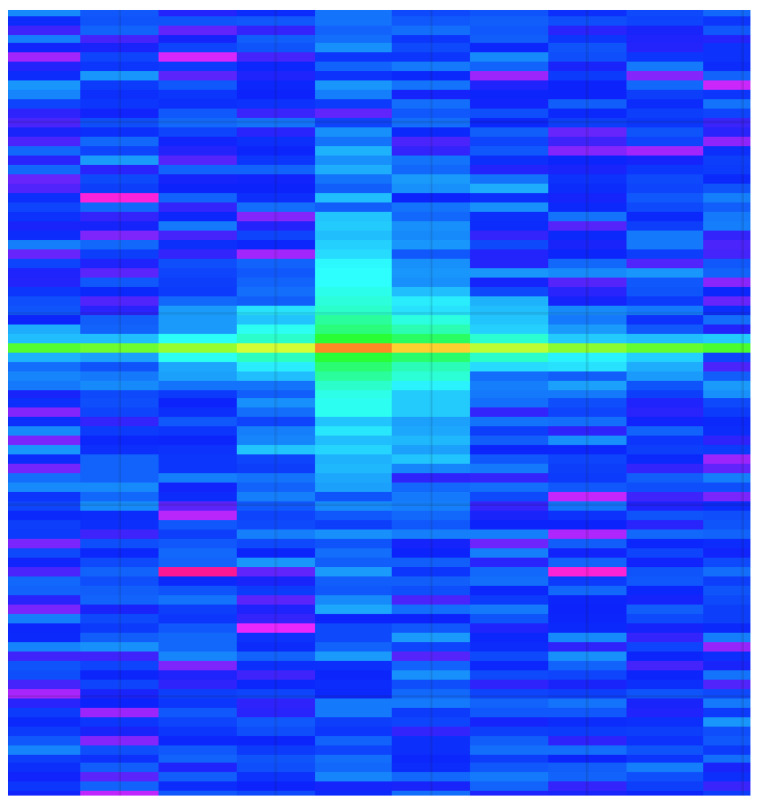
Generated radar range-Doppler heatmap example.

**Figure 6 sensors-22-08859-f006:**
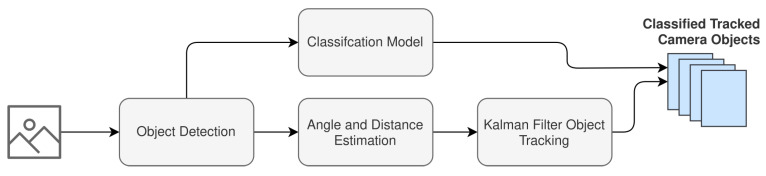
Camera processing pipeline design.

**Figure 7 sensors-22-08859-f007:**
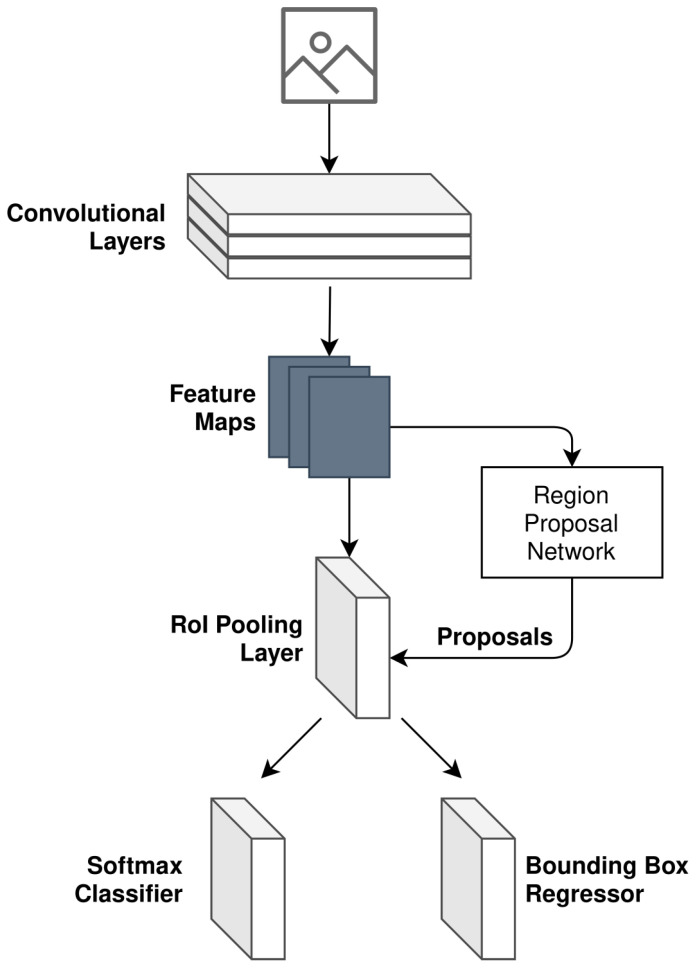
Faster R-CNN model design as used for camera object detection.

**Figure 8 sensors-22-08859-f008:**
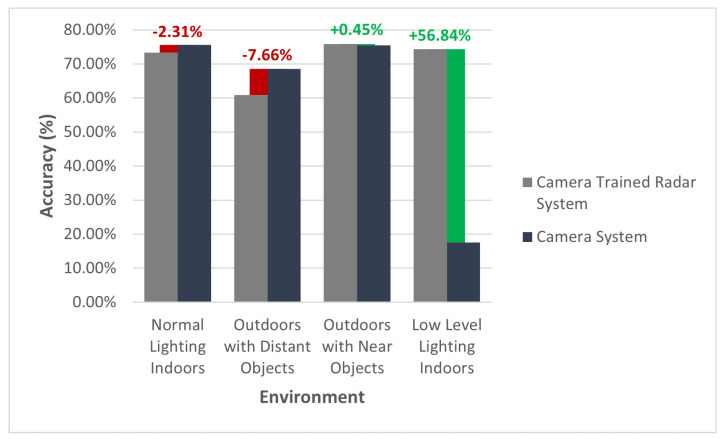
Camera-trained radar system accuracy in contrast to a trained standalone camera system.

**Figure 9 sensors-22-08859-f009:**
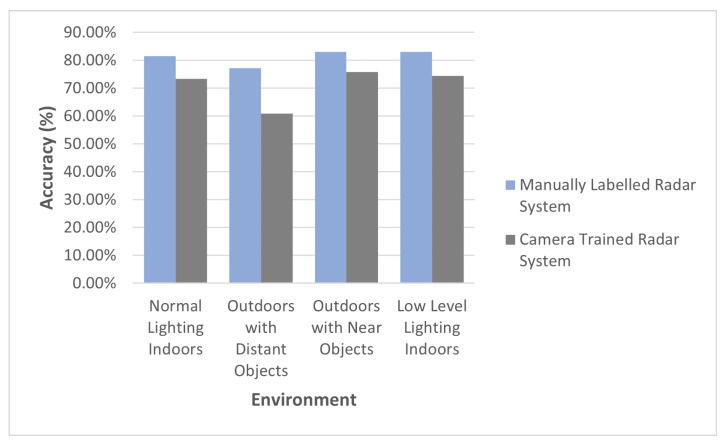
The manually labeled trained radar system accuracy in contrast to the camera-trained radar system.

**Table 1 sensors-22-08859-t001:** Types of mmWave radar and camera fusion system architectures.

Architecture Type	Description	Benefits & Limitations
Centralized	This refers to an architecture where the individual raw data of both the camera and mmWave radar is obtained independently and converged in a central processor for processing.	Benefits: Low information loss, original data preserved, simple structure, and a high processing rate.Limitations: Independent sensor units, large communication bandwidth required, high computing power needed by a centralized unit, and a single point of failure.
Distributed	This refers to an approach where each the radar and camera process their own data independently and send the post-processed data to a central fusion unit to then perform fusion on the post-processed data.	Benefits: Reducing the transmission time, reduced pressure on the fusion center, higher reliability resistance, and low communication bandwidth.Limitations: Data collection units also require the capability of processing the data, and the central processor is operating on post-processed data resulting in reduced flexibility.
Hybrid	The hybrid fusion approach refers to an architecture where some sensors follow the centralized approach, as defined above, and others follow the distributed approach, also as defined above. Measurements from all sensors are combined into a hybrid measurement, which in turn, is used to update the final data.	Benefits: Advantages of both centralized and distributed are retained as well as flexibility in satisfying varying requirements.Limitations: Complex data structures, increased computational and communication load, and high design requirements.

**Table 2 sensors-22-08859-t002:** Types of mmWave radar and camera-fusion depths.

Fusion Depth	Description
Low level	This class of fusion depth is best considered to be at the data level. It refers to a level of fusion that takes the raw data from each sensor to form a synthetic dataset illustrating a raw fused state, ready to be further processed.
Medium level	This refers to a class of fusion that takes place once several primitive features have been derived for each sensor independently and are fused to form a feature super set.
High level	This fusion level is considered an advanced form of fusion. Fusion at this level takes place once independent outcomes have been derived for each sensor, and the fused result is an expression of the combined sensor specific outcomes.

**Table 3 sensors-22-08859-t003:** Distribution of activities in the recorded dataset.

Activity	Distribution
Running	26.69%
Walking	25.02%
Falling	23.34%
Unknown	24.95%

**Table 4 sensors-22-08859-t004:** Accuracy similarity between the standalone camera system and camera-trained radar system.

Environment	Trained Similarity
Normal lighting indoors	97.69% ↓
Outdoors with distant objects	92.34% ↓
Outdoors with near objects	99.55% ↑
Low level lighting indoors	43.16% ↑

## Data Availability

Not applicable.

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
