# Peer review of "A Combined mmWave Tracking and Classification Framework Using a Camera for Labeling and Supervised Learning"

_sensors, 2022, doi:10.3390/s22228859_

Round 1

Reviewer 1 Report

The paper proposed a combined mmWave tracking and classification framework using camera for labeling and supervised. The paper is overall well-written but the demonstrated result was insufficient to validate the effectiveness of the proposed method. Details can be found below. 

1) As mentioned above, regardless of a lengthy paper, demonstrated results were quite a few summarized in Table 4. Furthermore, results shown in this Table is not sufficient to validate the effectiveness/usefulness of the proposed novel framework. Compared to conventional method, the improvement of the proposed one is quite negligible, except for the low lighting scenario. In some other scenarios like outdoor with distant objects, the performance even degraded. 

2) The reviewer expected that the authors should conduct more validation experiments to demonstrate the effectiveness of the proposed framework in the future. Also, one more comparison scheme i.e. mmWave radar alone (without camera information fusion) should be added to truly show the benefits of the proposed method. 

3) In Eq. (5), the dot symbol is used. The reviewer believed that it means "multiplication" but there was no explanation in this paper. Also, for other formulae related to "multiplication", the same marks were not reused. Please unify representation of the paper. 

Reviewer 2 Report

The authors propose a framework that uses cameras to assist in labeling tags of millimeter-wave radar, which reduces the workload of manual labeling data. And the authors collect corresponding data for actual scenes, which verifies the effectiveness of the proposed framework. The specific amendments are as follows:

1. Serial numbers (1. 2. 3.) in section 2.1.2 on page 6 is recommended to replace because it is confused with the chapter title. There is a similar issue with the “Results” location.

2. Figure. 2 is very blurry, so it is recommended to replace it to enhance the image clarity.

3. The reference format needs to be consistent, and some DOIs are missing.

4. How is the generalization performance of the proposed algorithm? Such as the effects of radar resolution, required sample number of training.

Reviewer 3 Report

The work reported and methodology adopted look interesting. Can authors share more details of the ADC used, such as type, resolution and sampling rate, etc.? 

Round 2

Reviewer 1 Report

The author did not resolve the comments of the reviewer properly. Especially for the 2nd comment. The reviewer requested the author to evaluate the radar-system alone to demonstrate the effectiveness of the proposed method (fusion of camera and radar data) but the author said that it is their future work. With the current evaluation results, the reviewer cannot judge the effectiveness of the proposed method from the newly added Figure 8. In this figure, it appears to the reviewer that the proposed method is only effective for certain scenario like low-light condition. Such condition is advantageous for radar system. Therefore, it is not clear to the reviewer if the improvement is due to mmWave radar merely or owing the the fusion of the camera data and radar data (the proposed method). 
